# Point-Substitution of Phenylalanine Residues of 26RFa Neuropeptide: A Structure-Activity Relationship Study

**DOI:** 10.3390/molecules26144312

**Published:** 2021-07-16

**Authors:** Benjamin Lefranc, Karima Alim, Cindy Neveu, Olivier Le Marec, Christophe Dubessy, Jean A. Boutin, Julien Chuquet, David Vaudry, Gaëtan Prévost, Marie Picot, Hubert Vaudry, Nicolas Chartrel, Jérôme Leprince

**Affiliations:** 1INSERM U1239, Laboratory of Neuronal and Neuroendocrine Differentiation and Communication, Normandy University, 76000 Rouen, France; benjamin.lefranc@univ-rouen.fr (B.L.); alimkarima8@gmail.com (K.A.); duchemin@neurofit.com (C.N.); lemarec.olivier@gmail.com (O.L.M.); christophe.dubessy@univ-rouen.fr (C.D.); julien.chuquet@univ-rouen.fr (J.C.); david.vaudry@univ-rouen.fr (D.V.); gaetan.prevost@chu-rouen.fr (G.P.); marie.picot@univ-rouen.fr (M.P.); hubert.vaudry@univ-rouen.fr (H.V.); nicolas.chartrel@univ-rouen.fr (N.C.); 2Cell Imaging Platform of Normandy, PRIMACEN, Normandy University, 76000 Rouen, France; 3Institut de Recherches Internationales SERVIER, 50 rue Carnot, 92284 Suresnes, France; ja.boutin.pro@gmail.com; 4PHARMADEV, Faculté de Pharmacie, Université de Toulouse, 31062 Toulouse, France

**Keywords:** GPR103, QRFP, RFamide, intracellular calcium concentration, peptide analog

## Abstract

26RFa is a neuropeptide that activates the rhodopsin-like G protein-coupled receptor QRFPR/GPR103. This peptidergic system is involved in the regulation of a wide array of physiological processes including feeding behavior and glucose homeostasis. Herein, the pharmacological profile of a homogenous library of QRFPR-targeting peptide derivatives was investigated in vitro on human QRFPR-transfected cells with the aim to provide possible insights into the structural determinants of the Phe residues to govern receptor activation. Our work advocates to include in next generations of 26RFa_(20–26)_-based QRFPR agonists effective substitutions for each Phe unit, i.e., replacement of the Phe^22^ residue by a constrained 1,2,3,4-tetrahydroisoquinoline-3-carboxylic acid moiety, and substitution of both Phe^24^ and Phe^26^ by their *para*-chloro counterpart. Taken as a whole, this study emphasizes that optimized modifications in the C-terminal part of 26RFa are mandatory to design selective and potent peptide agonists for human QRFPR.

## 1. Introduction

The neuropeptides 26RFa (QRFP26) and its N-terminal extended form 43RFa (QRFP, Figure 1) are the endogenous ligands of the G_i/s/q_ coupled QRFP receptor (QRFPR), formerly known as the orphan receptor GPR103 [1,2,3]. Since the isolation of 26RFa from a European green frog *Rana esculenta* brain extract [4], prepro-26RFa cDNA has been characterized in numerous genomes of diverse vertebrates from fish to mammals [5]. However, very few mature peptides have been identified so far, leaving the 26RFa/QRFP precursor post-translational processing uncertain. Indeed, in avians, 26RFa orthologs have been isolated and sequenced from the Japanese quail [6] and zebra finch [7] while, in rodents, only the 43RFa counterpart has been characterized from rat [2,8]. In humans, both 26RFa- and 43RFa-immunoreactive forms have been detected in hypothalamic and spinal cord extracts [9]. Interestingly, the presence of a tribasic cleavage site (Arg-Lys-Arg/Lys) within the 26RFa sequence suggests that the C-terminal heptapeptide GGFSFRF-NH_2_, named 26RFa_(20–26)_, could also be produced from the precursor. However, to date, this fragment, whose sequence is highly conserved in tetrapods, has not been identified as a mature product of the 26RFa/QRFP gene transcription.

In rats, the 26RFa/QRFP gene is primarily expressed in hypothalamic neurons which project in various brain areas [1,2,8,10]. Consistent with the widespread distribution of 26RFa/QRFP-containing fibers, 26RFa and/or QRFP are involved in the regulation of multiple physiological activities [5]. In particular, it is now firmly established that 26RFa and QRFP stimulate food intake in rodents [1,8,10,11,12,13,14,15,16]. There is also evidence that 26RFa and/or QRFP regulate several other neuroendocrine and cognitive functions including reproduction [17,18], anxiety [19], memory [20], cardiovascular activity [8] and nociceptive transmission [21,22,23], some of which depend, at least in part, on off-target interaction with QRFPR-related receptors [24,25]. In addition, recent data reveal that the 26RFa/QRFPR system controls glucose homeostasis at the periphery by increasing insulin sensitivity and inhibiting hepatic glucose production [26,27,28,29]. Thus, QRFPR positions as an attractive target for the development of innovative drugs. Indeed, there is now clinical evidence for possible indications of QRFPR ligands for the treatment of metabolic disorders, obesity and diabetes [5].

As a molecular signature, the RF-amide C-terminal extremity of 26RFa/QRFP and other RF-amide-related peptides (i.e., NPAF/NPFF, PrRP, RFRP-1(GnIH)/RFRP-3 and kisspeptin in humans [30]) represents the chemical determinant for bioactivity. Notably, deamidation or local side-chain substitution of the RF-amide moiety is generally associated with full or partial loss of activity and/or affinity of these neuropeptides [2,31,32,33]. The N-terminal part of RF-amide-related peptides diverges in their sequence and length, and conditions the whole or partial affinity and selectivity for one or several receptors, respectively [34,35]. For instance, 26RFa also binds with nanomolar affinity to both NPFF1 and NPFF2 receptors [24]. However, 26RFa can be downsized from the N-terminal extremity with a gradual decrease in its ability to mobilize intracellular calcium concentration ([Ca^2+^]_i_) in CHO cells transfected by the human QRFPR (*h*QRFPR) [36]. Since 26RFa_(20–26)_ (LV-2021) mimics the orexigenic and gonadotropic effects of full-length 26RFa [12,17], it appears that this C-terminal heptapeptide can serve as a molecular scaffold for designing low molecular weight, potent and stable *h*QRFPR ligands. As a matter of fact, 26RFa_(20–26)_ has already been successfully used to design nanomolar active agonists such as [Nva^23^]26RFa_(20–26)_ (LV-2073) [36], [Cmpi^21^, aza-β^3^-Hht^23^]26RFa_(21–26)_ (LV-2172) [37], and [(Me)^ω^Arg^25^]26RFa_(20–26)_ (LV-2186) [38], as well as proteolytic resistant pseudopeptides (Figure 2) [39]. According to our previous structure-activity relationship studies [for review, 5], these compounds exhibit modifications at positions occupied by residues classified as permissive to substitution or engaged in peptide bonds susceptible to degradation. In contrast, the residues of the -Phe^24^-Arg^25^-Phe^26^-NH_2_ triad are very sensitive to substitutions and only tolerate subtle amendments for improving bioactivity like the ω-methylation of arginine 25 [38]. To date, few replacements have been reported in the phenylalanine positions of 26RFa_(20–26)_ [5]. As previously suggested for PrRP and kisspeptin [40], our work emphasizes that optimized modifications in the C-terminal part of 26RFa are mandatory to design selective and potent peptidergic ligands for *h*QRFPR.

Thus, the aim of the present study was to investigate molecular diversity at the unexplored Phe positions to go deeper into the structure-activity relationships of 26RFa for the design of potent *h*QRFPR agonists.

## 2. Results and Discussion

### 2.1. Impact of Modifications of Each Phenylalanine Residue

It is well-accepted that the (hetero)aromatic units of peptides and proteins are master residues for DNA recognition [41], folding [42] and receptor-ligand interaction [43]. In particular, His, Phe, Trp and Tyr contribute to three types of non-covalent interactions including π-hydrogen bonds, electrostatic cation-π interactions and van der Waals π-π interactions [44] for governing molecular recognition of a ligand into the binding site of the receptor and transduction processes. 

Replacement of all three phenylalanine residues of 26RFa_(20–26)_ (LV-2021, **1**) with an alanine moiety identifies Phe^24^ and Phe^26^ as key residues for QRFPR activation, unlike Phe^22^ which is rather permissive to this substitution and thus weakly participates in the activity of the peptide [36]. Accordingly, Phe^24^ and Phe^26^ side chains strongly interact with hydrophobic regions at the bottom of the binding pocket of the *h*QRFPR homology model [45]. In order to optimize positions 22, 24 and 26 of LV-2021, we have successively replaced each phenylalanine of the heptapeptide **1** with different commercially available aromatic or aliphatic building blocks (Table 1, compounds **2**–**48**) and evaluated their ability to mobilize [Ca^2+^]_i_ in cultured G_α16_-*h*QRFPR-transfected CHO cells (Table 2), as previously described [36]. Substitution with the isosteric residue 3-(2-thienyl)-alanine (Thi) yielded compounds LV-2050, LV-2051 and LV-2052 (**2**–**4**) that were less active than the parent peptide whatever the position concerned (Table 2). Similarly, incorporation of more hindered residues such as 3-(2-naphtyl)-alanine (2Nal; LV-2065, LV-2213, LV-2191, **5**–**7**) and tryptophane (LV-2210, LV-2204, LV-2187, **8**–**10**) altered the agonistic activity of the compounds, suggesting that there was little space available at the bottom of the orthosteric binding site as previously reported [45]. However, the para position of the Phe^26^ residue tolerated a bulky group or a substituent with different electronic effects. Indeed, [*pt*BuPhe^26^]26RFa_(20–26)_ (LV-2238, **13**), [Pcp^26^]26RFa_(20–26)_ (LV-2193, **16**) and [*p*NO_2_Phe^26^]26RFa_(20–26)_ (LV-2194, **19**) were significantly more potent than 26RFa_(20–26)_ (LV-2021, **1**) to increase [Ca^2+^]_i_ in vitro (Table 2, Figure 3A–C). The same substitutions at positions Phe^22^ and Phe^24^ were either neutral (**15**, **17**) or deleterious to activity (**11**, **12**, **14**, **18**). Finally, methylene shortening of the phenylalanine side chain led to the inactive phenyl-glycine (Phg)-containing [Phg^22^]26RFa_(20–26)_ (LV-2053, **20**), [Phg^24^]26RFa_(20–26)_ (LV-2054, **21**; LV-2055, **22**) and [Phg^26^]26RFa_(20–26)_ (LV-2056, **23**; LV-2057, **24**) diastereoisomers, probably due to the lack of interactions with hydrophobic residues in the core of the *h*QRFPR binding pocket.

The heptapeptide 26RFa_(20–26)_ does not encompass any secondary amide bonds within its backbone. Thus, nine analogs with *N*-methyl-phenylalanine (*N*MePhe), 1,2,3,4-tetrahydroisoquinoline-3-carboxylic acid (Tic) or *N*-benzyl-glycine (*N*Phe) were synthesized. These modifications were aimed at analyzing the significance of H-bond contributions of the amide NHs. Alkylation of these NHs without loss of agonistic activity would facilitate further analog design. *N*-methylation of Phe^22^ (LV-2066, **25**) and Phe^26^ (LV-2242, **27**) nullified the activity of the parent compound (EC_50_ > 10 µM, Table 2) revealing the critical role of the corresponding NHs in forming an H-bond with the *h*QRFPR pocket residues or another residue of the peptide, whereas *N*-methylation of Phe^24^ (LV-2233, **26**) did not impair the whole activity of LV-2021 (**1**). The corresponding Tic-containing analogs (**28**–**30**) did not totally confirm these data since [Tic^24^]26RFa_(20–26)_ (LV-2205, **29**) and [Tic^26^]26RFa_(20–26)_ (LV-2189, **30**) were devoid of effect while [Tic^22^]26RFa_(20–26)_ (LV-2211, **28**) was 5-fold more potent than 26RFa_(20–26)_ (LV-2021, **1**) (Figure 3D). It is thus also plausible that the structural constraint induced by the Tic^22^ residue may keep the side chain in a favorable conformation for receptor activation, which was less accessible by free rotation in the native Phe side chain. The *N*Phe-peptoid analogs (**31**–**33**) behaved like the Tic-containing counterparts, and [*N*Phe^22^]26RFa_(20–26)_ (LV-2058, **31**) was almost as potent as LV-2211 (Table 2, Figure 3E). At this stage, conclusions are compromised since the local conformational rigidity to peptide backbone via pipecolinic acid bridge [46] and the shift of side chain functionality from the α-carbon to the NH offering flexibility to peptoid led to similar results. However, we can speculate that the Phe^22^ residue, which does not dive deeply into the binding pocket, accommodated rather well with backbone modifications at the odds to Phe^24^ and Phe^26^ which are strongly embedded in the orthosteric binding site [45]. An increase in H-bond capacity and flexibility by the introduction of (3*S*,4*S*)-4-amino-3-hydroxy-5-phenylpentanoic acid (AHPPA), a marine *Cyanobacterium symploca* peptide naturally occurring γ-amino acid [47] (**34**–**36**) had no positive effect on the potency of the parent heptapeptide (Table 2).

We have previously reported that stereoinversion to D-Phe in [DPhe^22^]26RFa_(20–26)_, [DPhe^24^]26RFa_(20–26)_ and [DPhe^26^]26RFa_(20–26)_ results in a complete loss of the [Ca^2+^]_i_ response [36]. Similarly, successive D-Trp and D-Tic incorporation within the 26RFa_(20–26)_ sequence in the same positions generated weak (LV-2237, **37**; LV-2212, **40**; LV-2215, **41**) or inactive (LV-2070, **38**; LV-2188, **39**; LV-2190, **42**) agonists (Table 2), confirming that not only side chain functionality but also its correct orientation play a critical role in the activity of the peptide.

Systematic replacement of the three Phe units with the constrained aliphatic residue octahydroindole-2-carboxylic acid (Oic) yielded LV-2067 (**43**), LV-2234 (**44**) and LV-2241 (**45**) that were totally devoid of Ca^2+^-mobilizing activity (Table 2). The Oic moiety might induce a turn, like the prolyl residue [48], that would destabilize at this point the correct folding of the peptide backbone in the *h*QRFPR binding pocket. Furthermore, these results also confirm the importance of aromaticity in positions 22, 24 and 26 of 26RFa_(20–26)_ in the receptor activation process.

In methanol, 26RFa adopts a well-defined conformation consisting of a N-terminal amphipathic α-helical structure (Pro^4^–Arg^17^), preceding a C-terminal disordered region [49]. Interestingly, it has been shown that two Phe residues in an *i*, *i+4* arrangement (Phe^22^ and Phe^26^ of 26RFa and 26RFa_(20–26)_) enthalpically stabilize an α-helix [50]. To assess such a conformation of the C-terminal extremity of the peptide inside the *h*QRFPR binding pocket during the recognition process, we prepared cyclic disulfide-bridged 26RFa_(20–26)_ analogs (Table 1, **46**–**48**) as already reported for the pentapeptide Met-enkephalin [51]. The activity profiles of [Cys^22,26^]26RFa_(20–26)_ (LV-2107, **47**), [Cys^22,24^]26RFa_(20–26)_ (LV-2105, **46**) and [Cys^24,26^]26RFa_(20–26)_ (LV-2106, **48**) did not confirm that the Phe residues of 26RFa_(20–26)_ (LV-2021, **1**) participate in the bioactive conformation of the peptide or that some mandatory features are absent of these analogs (Table 2).

Compounds inactive as agonists were evaluated as possible antagonists of 26RFa-evoked calcium increase in cultured G_α16_-*h*QRFPR-transfected CHO cells. None of the members of this series were able to reverse the stimulatory effect of 10^−7^ M 26RFa on [Ca^2+^]_i_, except LV-2068 (**11**) and LV-2188 (**39**) which antagonized 45–46% of the agonistic response with, respectively, modest and very low IC_50_ (Table 2, Figure 4).

To summarize, the three phenylalanine units exhibit differential contributions to the biological activity of 26RFa_(20–26)_. The Phe^22^ residue appears to be the most permissive as it tolerated *N*-substitution like in [Tic^22^]26RFa_(20–26)_ (LV-2211, **28**) which was 5-folds more potent than the lead peptide. Conversely, most modifications of Phe^24^ and Phe^26^ did not improve the activity with the exception of para substituents of lesser size than that of a *tert*butyl group. Indeed, [Pcp^24^]26RFa_(20–26)_ (LV-2232, **15**), [Pcp^26^]26RFa_(20–26)_ (LV-2193, **16**) and [*p*NO_2_Phe^26^]26RFa_(20–26)_ (LV-2194, **19**) were 2-, 3.5- and 3.3-fold more potent than 26RFa_(20–26)_ (LV-2021, **1**). Although the amide NH of Phe^24^ residue was probably not involved in H-bond, our results suggest that some flexibility of the peptide backbone is required at this point. Noteworthy, not only the H-bond capacity, but other features were also impacted by these modifications such as the side chain presentation geometry, amide cis/trans isomerization equilibrium, and/or β-sheet potential of the analog with a wide range of steric, electronic and hydrophobic characteristics.

### 2.2. Impact of Concomitant Modifications of Gly^20^, Gly^21^ and Phe^22^ Residues

The variety of conformations characterized in enkephalins [52] and other short peptide models containing contiguous Gly units [53] reveals an important heterogeneity in 3D structures of the Gly-Gly segment influenced by both neighboring residues and environment. In the homology model of *h*QRFPR, the Gly-Gly motif of 26RFa_(19–26)_ seems to adopt an extended conformation [45] generally observed in β-strands that form part of β-sheets. We have previously investigated the local requirement of the N-terminal dipeptide extremity of 26RFa_(20–26)_ and one of the most interesting results has been achieved by introducing a 4-carboxymethylpiperazine (Cmpi) unit in place of the native Gly-Gly pair which leads to the 5-fold more potent analog [Cmpi^21^]26RFa_(21–26)_ (LV-2043) than the reference heptapeptide [37]. Herein, to probe the space around this peptidomimetic moiety, we have introduced 2-piperazino-2-aryl-acetic acid racemate derivatives (Figure 5) in place of the first three 26RFa_(20–26)_ residues (Table 3) and evaluated the in vitro activity of each diastereoisomer (Table 4). In this series, analogs LV-2102 (**52**) and LV-2175 (**55**) containing 2-piperazino-2-[(4-fluoro)phenyl]acetic acid (Cmpi(4-FPhg)) and 2-piperazino-2-[(2-fluoro)phenyl]acetic acid (Cmpi(2-FPhg)) respectively, demonstrated similar [Ca^2+^]_i_-mobilizing activity in *h*QRFPR-transfected cells compared to the lead peptide, while other derivatives exhibited slightly (LV-2101, **51**; LV-2103, **53**; LV-2104 **54**; LV-2183, **63**) or detrimentally lower potency (Table 4). Altogether, these results suggest that incorporation of an aryl substituent on the methylene of the Cmpi building block with the concomitant deletion of the Phe^22^ residues do neither improve the ability of [Cmpi^21^]26RFa_(21–26)_ to activate *h*QRFPR nor reverse the 26RFa-evoked effect.

## 3. Materials and Methods

### 3.1. Reagents

All Fmoc-amino acid residues and building blocks and O-benzotriazol-1-yl-*N*,*N*,*N’*,*N’*-tetramethyluronium hexafluorophosphate (HBTU) were purchased from Christof Senn Laboratories (Dielsdorf, Switzerland) or PolyPeptide (Strasbourg, France). Rink amide 4-methylbenzhydrylamine (MBHA) resin was from Novabiochem (Darmstadt, Germany) and *N*-methylpyrrolidone (NMP), dimethylformamide (DMF) and dichloromethane (DCM) were from Biosolve (Dieuze, France). *N*,*N*-Diisopropylethylamine (DIEA), piperidine, trifluoroacetic acid (TFA), triisopropylsilane (TIS), *tert*-butylmethylether (TBME) were supplied from Sigma-Aldrich (Saint-Quentin-Fallavier, France). Acetonitrile was from Fisher Scientific (Illkirch, France) and α-cyano-4-hydroxycinnamic acid (CHCA) matrix from LaserBioLabs (Valbonne, France).

### 3.2. Peptide Synthesis and Purification

All peptides and derivatives were synthesized by the solid phase methodology on Rink amide MBHA resin using a Liberty microwave-assisted automated peptide synthesizer (CEM, Saclay, France) and the standard manufacturer’s procedures at 0.1 mmol scale as previously described [54]. All Fmoc-amino acids and building blocks (0.5 mmol, 5 equiv) were coupled by in situ activation with HBTU (0.5 mmol, 5 equiv) and DIEA (1 mmol, 10 equiv). Peptides and derivatives were deprotected and cleaved from the resin by adding a TFA/TIS/H_2_O (99.5/0.25/0.25) mixture for 120 min at room temperature. After filtration, crude peptides were precipitated by addition of TBME, centrifuged and recovered by elimination of the supernatant (3 folds). Peptides and derivatives were purified by reversed-phase HPLC (RP-HPLC) on a 2.2 × 25 cm Vydac 218TP1022 C_18_ column (Grace, Epernon, France) using a linear gradient (10–40%, 10–50%, 10–60%, 20–40%, 20–50% or 20–60% over 45 min) of acetonitrile/TFA (99.9/0.1) at a flow rate of 10 mL/min. The purified peptides were then characterized by MALDI-TOF mass spectrometry on a UltrafleXtreme (Bruker, Strasbourg, France) using CHCA as a matrix. Analytical RP-HPLC, performed on a 0.46 × 25 cm Vydac 218TP54 C_18_ column (Grace), showed that the purity of all compounds was >97.3%. 

### 3.3. Cell Culture

Stably transfected *h*QRFPR CHO cells were obtained as previously described [36,37,38]. The cells were maintained in F-12 nutrient mixture (Ham-F12) medium supplemented with 10% fetal bovine serum, 2 mM glutamine, and penicillin-streptomycin. Expression of G_α16_ cells was maintained using selection antibiotic hygromycin B (200 µg/mL) and that of *h*QRFPR using geneticin G418 (500 µg/mL) (Life Technologies, Villebon-Sur-Yvette, France) in a humidified 5% CO_2_ atmosphere at 37 °C.

### 3.4. Calcium Mobilization Assays

Changes in intracellular Ca^2+^ concentrations induced by 26RFa_(20–26)_ analogs in CHO-G_α16_-*h*QRFPR-transfected cells were measured on a benchtop scanning fluorometer Flexstation III (Molecular Devices, Sunnyvale, CA, USA) as previously described [36,37,38,55,56]. Briefly, 96-well assay black plates with clear bottoms (Corning international, Avon, France) were seeded at a density of 40,000 cells/well 24 h prior to assay. For profiling agonistic experiments, cells were loaded with 2 µM Fluo-4 acetoxymethyl ester (AM) (Invitrogen) for 1 h in the presence of 0.01% pluronic acid, washed thrice, and incubated for 30 min with standard HBSS containing 2.5 mM probenecid and 5 mM HEPES. Compounds to be tested were added at final concentrations ranging from 10^−11^ to 10^−5^ M in HBSS, and the fluorescence intensity was measured during 3 min. To evaluate the antagonistic potency of the test compounds, cells were incubated with each compound over 15 min after Fluo-4 AM loading. Then, during fluorescence recording, a pulse of 26RFa was administered at a final concentration of 10^−7^ M. After subtraction of the mean fluorescence background, the baseline was normalized to 100%. Fluorescence peak values were determined for each concentration of compound.

### 3.5. Statistical Analysis

Calcium experiments were performed in triplicate, and data, expressed as mean ± SEM of at least three distinct experiments, were analyzed with the Prism 8.0 software (Graphpad Software, San Diego, CA, USA). EC_50_ and the IC_50_ values were determined from concentration–response curves using a sigmoidal dose–response fit with variable slope from at least three independent determinations. Differences between 26RFa_(20–26)_ and analog activities were analyzed by the Mann–Whitney test. *p* values < 0.05 were considered significant.

### 3.6. Nomenclature of Targets and Ligands

All targets and ligands used throughout this manuscript conform with the guidelines outlined by the International Union of Basic and Clinical Pharmacology and British Pharmacological Society (IUPHAR/BPS) Guide to Pharmacology [5,57].

## 4. Conclusions

The C-terminal extremity of 26RFa was previously identified as a pivotal region to modulate its signaling at *h*QRFPR [36]. In this work, the Phe^22^, Phe^24^ and Phe^26^ residues of 26RFa_(20–26)_ were modified using series of point substitutions with natural, side chain-constrained, side chain-modified residues and peptidomimetic building blocks. Subtle chemical modifications in the sequence led to significant improvement in compound potency to activate *h*QRFPR. As such, a single modification of Phe^22^ with the steric restricted Tic residue decreased the EC_50_ value from 1640 ± 259 to 327 ± 170 nM, providing the most efficient modification at this sequence position. Anchored substituents in the para position of each Phe unit were the most versatile investigated modification. Thereby, the introduction of a para-chloro-phenylalanine in place of the native Phe^24^ moiety emerged as a favorable replacement. This finding follows the same trend observed in [Pcp^26^]26RFa_(20–26)_, highlighting a limited room to accommodate aromatic residues around the Phe aryl group. Although combination of multiple point-effective modifications does not necessarily translate into an additive or synergic effect, we have explored each position of 26RFa_(20–26)_ for complete optimization of its sequence. Future challenges will be to convert several of these point modifications to low molecular weight 26RFa analogs for metabolic disorder, obesity or diabetes therapies. We are confident that, by utilizing subtle amendments, we can design 26RFa_(20–26)_-based compounds with nanomolar potency, functional selectivity and in vivo bioavailability.

## Figures and Tables

**Figure 1 molecules-26-04312-f001:**
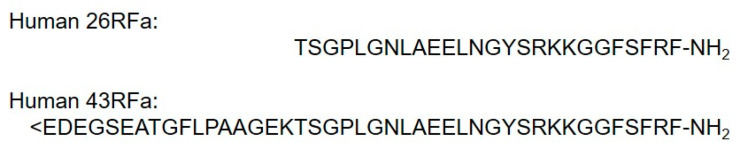
Primary structures of human 26RFa and 43RFa/QRFP. <E denotes pyroglutamic acid.

**Figure 2 molecules-26-04312-f002:**
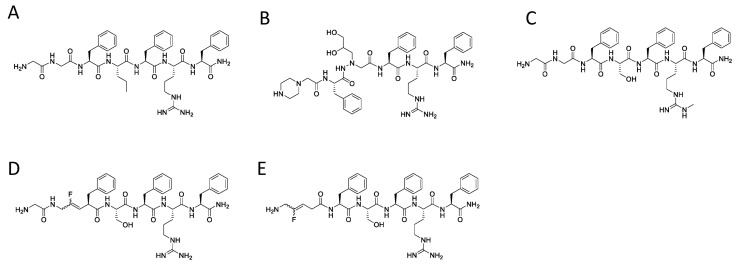
Chemical structures of peptidic and pseudopeptidic human QRFPR ligands. (**A**) [Nva^23^]26RFa_(20–26)_ (LV-2073); (**B**) [Cmpi^21^, aza-β^3^-Hht^23^]26RFa_(21–26)_ (LV-2172); (**C**) [(Me)^ω^Arg^25^]26RFa_(20–26)_ (LV-2186); (**D**) [*ψ*(CF=CH)^21,22^]26RFa_(20–26)_; (**E**) [*ψ*(CF=CH)^20,21^]26RFa_(20–26)_.

**Figure 3 molecules-26-04312-f003:**
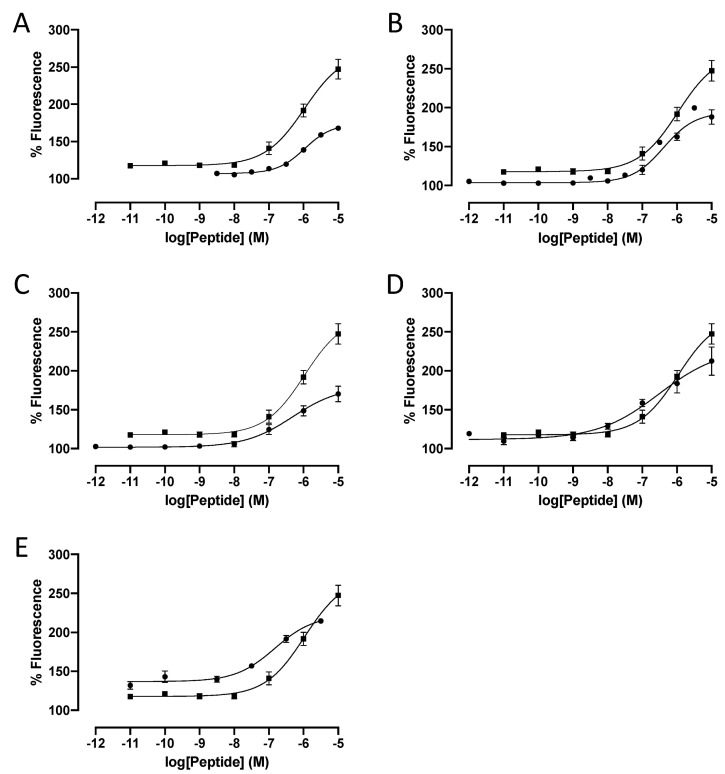
Effect of graded concentrations of Phe-modified 26RFa_(20−26)_ analogs on basal [Ca^2+^]_i_ mobilization in G_α16_-*h*QRFPR-transfected CHO cells. Prototype dose–response curves of 26RFa_(20−26)_ (LV-2021, **1**, closed square, (**A**–**E**)) and its analogs (closed circles) [*pt*BuPhe^26^]26RFa_(20–26)_ (LV-2238, **13**, (**A**)), [Pcp^26^]26RFa_(20–26)_ (LV-2193, **16**, (**B**)), [*p*NO_2_Phe^26^]26RFa_(20–26)_ (LV-2194, **19**, (**C**)), [Tic^22^]26RFa_(20–26)_ (LV-2211, **28**, (**D**)) and [*N*Phe^22^]26RFa_(20–26)_ (LV-2058, **31**, (**E**)). Data are mean ± SEM of triplicate. The EC_50_ calculated from these representative dose–response curves were 1069 nM for **1** (LV-2021), 1076 nM for **13** (LV-2238), 416 nM for **16** (LV-2193), 489 nM for **19** (LV-2194), 305 nM for **28** (LV-2211) and 149 nM for **31** (LV-2058).

**Figure 4 molecules-26-04312-f004:**
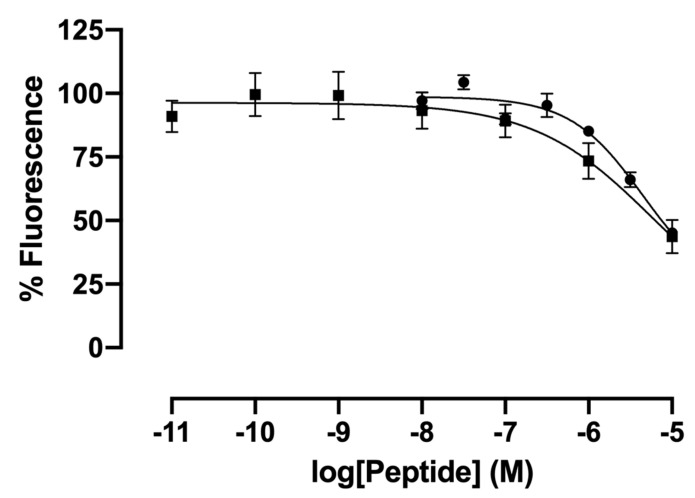
Effect of graded concentrations of Phe-modified 26RFa_(20−26)_ analogs on 26RFa-evoked [Ca^2+^]_i_ mobilization in G_α16_-*h*QRFPR-transfected CHO cells. Prototype dose-inhibition curves of [*pt*BuPhe^22^]26RFa_(20–26)_ (LV-2068, **11**, closed circles) and [DTrp^26^]26RFa_(20–26)_ (LV-2188, **39**, closed squares) on 10^−7^ M 26RFa-evoked response. Data are mean ± SEM of triplicate. The IC_50_ value calculated from these representative dose–inhibition curves were 4495 nM for **11** (LV-2068) and 5841 nM for **39** (LV-2188).

**Figure 5 molecules-26-04312-f005:**
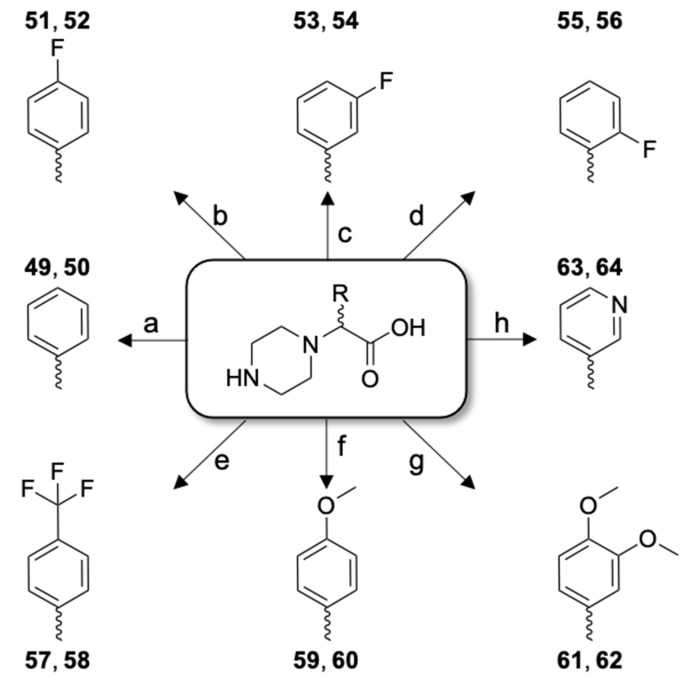
2-Piperazino-2-aryl-acetic acid derivatives used in place of the H-Gly-Gly-Phe- sequence and incorporated in **49**–**64**. These peptidomimetic moieties were used as racemate.

**Table 1 molecules-26-04312-t001:** Chemical data for 26RFa_(20–26)_ analogs substituted in positions 22, 24 and 26.

	Peptide	Structure of Residue 22, 24, 26	Code	HPLC	MS
	Rt (min) ^a^	Purity (%)	Calcd ^b^	Obsd ^c^
**1**	26RFa_(20–26)_		LV-2021	18.0	99.9	815.41	816.53
**2**	[Thi^22^]26RFa_(20–26)_	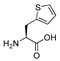	LV-2050	18.6	99.9	821.36	822.33
**3**	[Thi^24^]26RFa_(20–26)_	LV-2051	18.6	99.9	821.36	822.37
**4**	[Thi^26^]26RFa_(20–26)_	LV-2052	18.6	99.9	821.36	822.38
**5**	[2Nal^22^]26RFa_(20–26)_	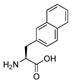	LV-2065	20.4	99.9	865.42	866.45
**6**	[2Nal^24^]26RFa_(20–26)_	LV-2213	20.8	99.9	865.42	866.46
**7**	[2Nal^26^]26RFa_(20–26)_	LV-2191	20.6	99.9	865.42	866.57
**8**	[Trp^22^]26RFa_(20–26)_	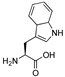	LV-2210	18.9	99.9	854.42	855.27
**9**	[Trp^24^]26RFa_(20–26)_	LV-2204	18.6	99.9	854.42	855.43
**10**	[Trp^26^]26RFa_(20–26)_	LV-2187	19.3	99.9	854.42	855.48
**11**	[*pt*BuPhe^22^]26RFa_(20–26)_	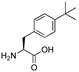	LV-2068	22.1	99.9	871.47	872.49
**12**	[*pt*BuPhe^24^]26RFa_(20–26)_	LV-2235	22.3	99.9	871.47	872.45
**13**	[*pt*BuPhe^26^]26RFa_(20–26)_	LV-2238	22.4	99.9	871.47	872.55
**14**	[Pcp^22^]26RFa_(20–26)_	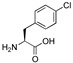	LV-2069	19.9	99.9	849.37	850.30
**15**	[Pcp^24^]26RFa_(20–26)_	LV-2232	20.5	99.9	849.37	850.41
**16**	[Pcp^26^]26RFa_(20–26)_	LV-2193	20.0	99.9	849.37	850.42
**17**	[*p*NO_2_Phe^22^]26RFa_(20–26)_	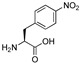	LV-2214	19.9	99.9	860.39	861.53
**18**	[*p*NO_2_Phe^24^]26RFa_(20–26)_	LV-2236	19.0	99.9	860.39	861.46
**19**	[*p*NO_2_Phe^26^]26RFa_(20–26)_	LV-2194	18.9	99.9	860.39	861.56
**20**	[Phg^22^]26RFa_(20–26)_ *dia 2*	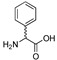	LV-2053	24.2	97.8	801.39	802.40
**21**	[Phg^24^]26RFa_(20–26)_ *dia 1*	LV-2054	21.1	99.9	801.39	802.40
**22**	[Phg^24^]26RFa_(20–26)_ *dia 2*	LV-2055	22.2	98.3	801.39	802.40
**23**	[Phg^26^]26RFa_(20–26)_ *dia 1*	LV-2056	21.0	99.9	801.39	802.70
**24**	[Phg^26^]26RFa_(20–26)_ *dia 2*	LV-2057	21.6	97.3	801.39	802.62
**25**	[*N*MePhe^22^]26RFa_(20–26)_	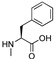	LV-2066	18.9	99.9	829.42	830.53
**26**	[*N*MePhe^24^]26RFa_(20–26)_	LV-2233	19.2	98.9	829.42	830.48
**27**	[*N*MePhe^26^]26RFa_(20–26)_	LV-2242	18.8	99.9	829.42	830.48
**28**	[Tic^22^]26RFa_(20–26)_	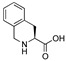	LV-2211	20.3	99.9	827.41	828.42
**29**	[Tic^24^]26RFa_(20–26)_	LV-2205	17.7	99.9	827.41	828.46
**30**	[Tic^26^]26RFa_(20–26)_	LV-2189	18.5	99.9	827.41	828.53
**31**	[*N*Phe^22^]26RFa_(20–26)_	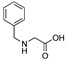	LV-2058	21.1	99.9	815.41	816.23
**32**	[*N*Phe^24^]26RFa_(20–26)_	LV-2060	17.9	99.9	815.41	816.56
**33**	[*N*Phe^26^]26RFa_(20–26)_	LV-2061	18.6	99.9	815.41	816.40
**34**	[AHPPA^22^]26RFa_(20–26)_	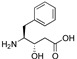	LV-2062	17.8	99.9	859.43	860.30
**35**	[AHPPA^24^]26RFa_(20–26)_	LV-2063	17.7	99.9	859.43	860.40
**36**	[AHPPA^26^]26RFa_(20–26)_	LV-2064	18.1	99.9	859.43	860.55
**37**	[DTrp^22^]26RFa_(20–26)_	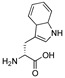	LV-2237	18.2	99.9	854.42	855.56
**38**	[DTrp^24^]26RFa_(20–26)_	LV-2070	17.7	99.9	854.42	855.53
**39**	[DTrp^26^]26RFa_(20–26)_	LV-2188	18.2	99.9	854.42	855.39
**40**	[DTic^22^]26RFa_(20–26)_	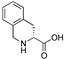	LV-2212	18.4	99.9	827.41	828.49
**41**	[DTic^24^]26RFa_(20–26)_	LV-2215	18.2	99.9	827.41	828.36
**42**	[DTic^26^]26RFa_(20–26)_	LV-2190	18.3	99.9	827.41	828.41
**43**	[Oic^22^]26RFa_(20–26)_	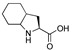	LV-2067	18.6	99.9	819.44	820.56
**44**	[Oic^24^]26RFa_(20–26)_	LV-2234	18.3	99.9	819.44	820.36
**45**	[Oic^26^]26RFa_(20–26)_	LV-2241	18.1	99.9	819.44	820.45
**46**	[Cys^22,24^]26RFa_(20–26)_	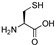	LV-2105	11.5	99.9	725.27	726.28
**47**	[Cys^22,26^]26RFa_(20–26)_	LV-2107	6.7	99.9	725.27	726.37
**48**	[Cys^24,26^]26RFa_(20–26)_	LV-2106	11.8	99.9	725.27	726.32

^a^ Retention time determined by analytical RP-HPLC. ^b^ Theorical monoisotopic molecular weight. ^c^
*m*/*z* [MH]^+^ value assessed by MALDI-TOF-MS. AHPPA, (3*S*,4*S*)-4-amino-3-hydroxy-5-phenylpentanoic acid; 2Nal, 3-(2-naphtyl)-alanine; *N*MePhe, *N*-methyl-phenylalanine; *N*Phe, *N*-benzyl-glycine; Oic, octahydroindole-2-carboxylic acid; Pcp, (4-chloro)-phenylalanine; Phg, phenylglycine; *p*NO_2_Phe, (4-nitro)-phenylalanine; *pt*BuPhe, (4-*ter*butyl)-phenylalanine; Thi, 3-(2-thienyl)-alanine; Tic, 1,2,3,4-tetrahydroisoquinoline-3-carboxylic acid. Diastereoisomers are numbered according to their elution order in RP-HPLC.

**Table 2 molecules-26-04312-t002:** Biological data for 26RFa_(20–26)_ analogs substituted in positions 22, 24 and 26.

	Peptide	Code	EC_50_			IC_50_			Imax
			(nM)			(nM)			(%) ^a^
**1**	26RFa_(20–26)_	LV-2021	1640	±	259		-		
**2**	[Thi^22^]26RFa_(20–26)_	LV-2050	3160	±	1110		-		
**3**	[Thi^24^]26RFa_(20–26)_	LV-2051	2530	±	1020		-		
**4**	[Thi^26^]26RFa_(20–26)_	LV-2052	7227	±	96		-		
**5**	[2Nal^22^]26RFa_(20–26)_	LV-2065	1150	±	170		-		
**6**	[2Nal^24^]26RFa_(20–26)_	LV-2213	15,300	±	5400		-		
**7**	[2Nal^26^]26RFa_(20–26)_	LV-2191	3790	±	1800		-		
**8**	[Trp^22^]26RFa_(20–26)_	LV-2210	1980	±	880		-		
**9**	[Trp^24^]26RFa_(20–26)_	LV-2204	2040	±	1600		-		
**10**	[Trp^26^]26RFa_(20–26)_	LV-2187	5060	±	1300		-		
**11**	[*pt*BuPhe^22^]26RFa_(20–26)_	LV-2068	>10,000			6220	±	2500	45
**12**	[*pt*BuPhe^24^]26RFa_(20–26)_	LV-2235	>10,000			ND			
**13**	[*pt*BuPhe^26^]26RFa_(20–26)_	LV-2238	1040	±	25 **		-		
**14**	[Pcp^22^]26RFa_(20–26)_	LV-2069	5000	±	4500		-		
**15**	[Pcp^24^]26RFa_(20–26)_	LV-2232	850	±	240 ^NS^		-		
**16**	[Pcp^26^]26RFa_(20–26)_	LV-2193	457	±	71 **		-		
**17**	[*p*NO_2_Phe^22^]26RFa_(20–26)_	LV-2214	2130	±	1100		-		
**18**	[*p*NO_2_Phe^24^]26RFa_(20–26)_	LV-2236	>10,000			ND			
**19**	[*p*NO_2_Phe^26^]26RFa_(20–26)_	LV-2194	491	±	33 **		-		
**20**	[Phg^22^]26RFa_(20–26)_ *dia 2*	LV-2053	5010	±	3600		-		
**21**	[Phg^24^]26RFa_(20–26)_ *dia 1*	LV-2054	>10,000			ND			
**22**	[Phg^24^]26RFa_(20–26)_ *dia 2*	LV-2055	>10,000			ND			
**23**	[Phg^26^]26RFa_(20–26)_ *dia 1*	LV-2056	>10,000			ND			
**24**	[Phg^26^]26RFa_(20–26)_ *dia 2*	LV-2057	>10,000			ND			
**25**	[*N*MePhe^22^]26RFa_(20–26)_	LV-2066	15,100	±	4200		-		
**26**	[*N*MePhe^24^]26RFa_(20–26)_	LV-2233	1360	±	750		-		
**27**	[*N*MePhe^26^]26RFa_(20–26)_	LV-2242	>10,000			ND			
**28**	[Tic^22^]26RFa_(20–26)_	LV-2211	327	±	170 *		-		
**29**	[Tic^24^]26RFa_(20–26)_	LV-2205	>10,000			ND			
**30**	[Tic^26^]26RFa_(20–26)_	LV-2189	7700	±	4400		-		
**31**	[*N*Phe^22^]26RFa_(20–26)_	LV-2058	578	±	110 *		-		
**32**	[*N*Phe^24^]26RFa_(20–26)_	LV-2060	>10,000			ND			
**33**	[*N*Phe^26^]26RFa_(20–26)_	LV-2061	>10,000			ND			
**34**	[AHPPA^22^]26RFa_(20–26)_	LV-2062	2850	±	380		-		
**35**	[AHPPA^24^]26RFa_(20–26)_	LV-2063	>10,000			ND			
**36**	[AHPPA^26^]26RFa_(20–26)_	LV-2064	>10,000			ND			
**37**	[DTrp^22^]26RFa_(20–26)_	LV-2237	6160	±	2400		-		
**38**	[DTrp^24^]26RFa_(20–26)_	LV-2070	>10,000			ND			
**39**	[DTrp^26^]26RFa_(20–26)_	LV-2188	>10,000			>10,000			46
**40**	[DTic^22^]26RFa_(20–26)_	LV-2212	4990	±	2100		-		
**41**	[DTic^24^]26RFa_(20–26)_	LV-2215	6200	±	2600		-		
**42**	[DTic^26^]26RFa_(20–26)_	LV-2190	>10,000			ND			
**43**	[Oic^22^]26RFa_(20–26)_	LV-2067	>10,000			ND			
**44**	[Oic^24^]26RFa_(20–26)_	LV-2234	>10,000			ND			
**45**	[Oic^26^]26RFa_(20–26)_	LV-2241	>10,000			ND			
**46**	[Cys^22,24^]26RFa_(20–26)_	LV-2105	>10,000			ND			
**47**	[Cys^22,26^]26RFa_(20–26)_	LV-2107	>10,000			ND			
**48**	[Cys^24,26^]26RFa_(20–26)_	LV-2106	>10,000			ND			

^a^ Inhibition at a maximal concentration of 10^−5^ M. Data are mean ± SEM of at least three independent experiments performed in triplicate. AHPPA, (3*S*,4*S*)-4-amino-3-hydroxy-5-phenylpentanoic acid; 2Nal, 3-(2-naphtyl)-alanine; *N*MePhe, *N*-methyl-phenylalanine; *N*Phe, *N*-benzyl-glycine; Oic, octahydroindole-2-carboxylic acid; Pcp, (4-chloro)-phenylalanine; Phg, phenylglycine; *p*NO_2_Phe, (4-nitro)-phenylalanine; *pt*BuPhe, (4-*ter*butyl)-phenylalanine; Thi, 3-(2-thienyl)-alanine; Tic, 1,2,3,4-tetrahydroisoquinoline-3-carboxylic acid. Diastereoisomers are numbered according to their elution order in RP-HPLC. ND, not detectable. NS, not significant. * *p* < 0.05, ** *p* < 0.01 vs. control 26RFa_(20–26)_ (LV-2021, **1**) as assessed by Mann and Whitney test.

**Table 3 molecules-26-04312-t003:** Chemical data for 26RFa_(22–26)_ analogs.

	Peptide Derivative	Code	HPLC		MS	
			Rt (min) ^a^	Purity (%)	Calcd ^b^	Obsd ^c^
**1**	26RFa_(20–26)_	LV-2021	18.0	99.9	815.41	816.53
**49**	[Cmpi(Phg)^22^]26RFa_(22–26)_ *dia 1*	LV-2099	12.8	99.9	756.41	757.46
**50**	[Cmpi(Phg)^22^]26RFa_(22–26)_ *dia 2*	LV-2100	12.9	99.9	756.41	757.46
**51**	[Cmpi(4-FPhg)^22^]26RFa_(22–26)_ *dia 1*	LV-2101	15.3	99.9	774.40	775.37
**52**	[Cmpi(4-FPhg)^22^]26RFa_(22–26)_ *dia 2*	LV-2102	15.7	99.9	774.40	775.39
**53**	[Cmpi(3-FPhg)^22^]26RFa_(22–26)_ *dia 1*	LV-2103	15.3	99.9	774.40	775.42
**54**	[Cmpi(3-FPhg)^22^]26RFa_(22–26)_ *dia 2*	LV-2104	15.8	99.9	774.40	775.41
**55**	[Cmpi(2-FPhg)^22^]26RFa_(22–26)_ *dia 1*	LV-2175	13.9	98.4	774.40	775.47
**56**	[Cmpi(2-FPhg)^22^]26RFa_(22–26)_ *dia 2*	LV-2176	14.7	99.9	774.40	775.53
**57**	[Cmpi(4-TfmPhg)^22^]26RFa_(22–26)_ *dia 1*	LV-2177	25.3	99.9	824.39	825.47
**58**	[Cmpi(4-TfmPhg)^22^]26RFa_(22–26)_ *dia 2*	LV-2178	26.2	99.9	824.39	825.39
**59**	[Cmpi(4-MeOPhg)^22^]26RFa_(22–26)_ *dia 1*	LV-2179	13.2	99.9	786.42	787.47
**60**	[Cmpi(4-MeOPhg)^22^]26RFa_(22–26)_ *dia 2*	LV-2180	13.4	99.9	786.42	787.41
**61**	[Cmpi(3,4-diMeOPhg)^22^]26RFa_(22–26)_ *dia 1*	LV-2181	10.9	99.9	816.43	817.37
**62**	[Cmpi(3,4-diMeOPhg)^22^]26RFa_(22–26)_ *dia 2*	LV-2182	11.5	99.9	816.43	817.49
**63**	[Cmpi(3-Pyg)^22^]26RFa_(22–26)_ *dia 1*	LV-2183	19.5	99.9	757.40	758.40
**64**	[Cmpi(3-Pyg)^22^]26RFa_(22–26)_ *dia 2*	LV-2184	20.3	99.9	757.40	758.31

^a^ Retention time determined by analytical RP-HPLC. ^b^ Theorical monoisotopic molecular weight. ^c^
*m*/*z* [MH]^+^ value assessed by MALDI-TOF-MS. Cmpi(Phg), 2-piperazino-2-phenylacetic acid; Cmpi(2-FPhg), 2-piperazino-2-[(2-fluoro)phenyl]acetic acid; Cmpi(3-FPhg), 2-piperazino-2-[(3-fluoro)phenyl]acetic acid; Cmpi(4-FPhg), 2-piperazino-2-[(4-fluoro)phenyl]acetic acid; Cmpi(4-TfmPhg), 2-piperazino-2-[(4-trifluoromethyl)phenyl]acetic acid; Cmpi(4-MeOPhg), 2-piperazino-2-[(4-methoxy)phenyl]acetic acid; Cmpi(3,4-diMeOPhg), 2-piperazino-2-[(3,4-dimethoxy)phenyl]acetic acid; Cmpi(3-Pyg), 2-piperazino-2-(3-pyridyl)acetic acid. Diastereoisomers are numbered according to their elution order in RP-HPLC.

**Table 4 molecules-26-04312-t004:** Biological data for 26RFa_(22–26)_ analogs.

	Peptide Derivative	Code	EC_50_			IC_50_
			(nM)			(nM)
**1**	26RFa_(20–26)_	LV-2021	1640	±	259	-
**49**	[Cmpi(Phg)^22^]26RFa_(22–26)_ *dia 1*	LV-2099	>10,000			ND
**50**	[Cmpi(Phg)^22^]26RFa_(22–26)_ *dia 2*	LV-2100	>10,000			ND
**51**	[Cmpi(4-FPhg)^22^]26RFa_(22–26)_ *dia 1*	LV-2101	4050	±	1200	-
**52**	[Cmpi(4-FPhg)^22^]26RFa_(22–26)_ *dia 2*	LV-2102	1660	±	788	-
**53**	[Cmpi(3-FPhg)^22^]26RFa_(22–26)_ *dia 1*	LV-2103	3560	±	440	-
**54**	[Cmpi(3-FPhg)^22^]26RFa_(22–26)_ *dia 2*	LV-2104	2710	±	1000	-
**55**	[Cmpi(2-FPhg)^22^]26RFa_(22–26)_ *dia 1*	LV-2175	2480	±	1113	-
**56**	[Cmpi(2-FPhg)^22^]26RFa_(22–26)_ *dia 2*	LV-2176	>10,000			ND
**57**	[Cmpi(4-TfmPhg)^22^]26RFa_(22–26)_ *dia 1*	LV-2177	>10,000			ND
**58**	[Cmpi(4-TfmPhg)^22^]26RFa_(22–26)_ *dia 2*	LV-2178	>10,000			ND
**59**	[Cmpi(4-MeOPhg)^22^]26RFa_(22–26)_ *dia 1*	LV-2179	>10,000			ND
**60**	[Cmpi(4-MeOPhg)^22^]26RFa_(22–26)_ *dia 2*	LV-2180	>10,000			ND
**61**	[Cmpi(3,4-diMeOPhg)^22^]26RFa_(22–26)_ *dia 1*	LV-2181	>10,000			ND
**62**	[Cmpi(3,4-diMeOPhg)^22^]26RFa_(22–26)_ *dia 2*	LV-2182	>10,000			ND
**63**	[Cmpi(3-Pyg)^22^]26RFa_(22–26)_ *dia 1*	LV-2183	4990	±	2800	-
**64**	[Cmpi(3-Pyg)^22^]26RFa_(22–26)_ *dia 2*	LV-2184	>10,000			ND

Data are mean ± SEM of at least three independent experiments performed in triplicate. Cmpi(Phg), 2-piperazino-2-phenylacetic acid; Cmpi(2-FPhg), 2-piperazino-2-[(2-fluoro)phenyl]acetic acid; Cmpi(3-FPhg), 2-piperazino-2-[(3-fluoro)phenyl]acetic acid; Cmpi(4-FPhg), 2-piperazino-2-[(4-fluoro)phenyl]acetic acid; Cmpi(4-TfmPhg), 2-piperazino-2-[(4-trifluoromethyl)phenyl]acetic acid; Cmpi(4-MeOPhg), 2-piperazino-2-[(4-methoxy)phenyl]acetic acid; Cmpi(3,4-diMeOPhg), 2-piperazino-2-[(3,4-dimethoxy)phenyl]acetic acid; Cmpi(3-Pyg), 2-piperazino-2-(3-pyridyl)acetic acid. Diastereoisomers are numbered according to their elution order in RP-HPLC. ND, not detectable.

## Data Availability

All data are generated during this study.

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
