# Peer review of "Point-Substitution of Phenylalanine Residues of 26RFa Neuropeptide: A Structure-Activity Relationship Study"

_molecules, 2021, doi:10.3390/molecules26144312_

Round 1
Reviewer 1 Report
In the present article, the authors described the impact of point-substitution of phenylalanine residues on the ability of 26RFa(20-26) to mobilize intracellular calcium. This paper is interesting and nicely written except some typos (Greek letter mostly) as well as some missing necessary background. Overall, I am not sure the paper can be publish in its current states sinceI feel that sometimes the authors have a little bit overinterpreted their data/results.
The introduction is informative and nicely written. However, I would have like a little more SAR background in order to better understand right from the beginning the rationale of modifying these phenylalanine residues. Also, a figure, recapitulating the different structure of the compounds cited page 2 line 75-76, would be interesting.
Despite referring to a publish article regarding the use of a Ga16-hQRFPR-transfected cell line, I still to not understand why a conventional this particular cell background was needed especially considering the low potency observed with the native ligand.
Also, I was really surprised that no binding affinity is reported. I understand that phenylalanine substitution being at the heart of the article, it was not possible to substitute one of them to introduce of Tyr moiety for subsequent iodination procedure but why not using other strategy to prepare a radioligand. By not investigating this parameter, the author could miss some important information regarding the recognition and activation process.
Most if not all of the presented concentration-response curves do not present a clear plateau. One curve, possibly panel D, looks more like a straight line and I do not see how a EC50 can be deduced from it. Also, some of the compounds (difficult to identify which one since the figure is no labelled according to the legend), seem to be better or worse than the reference to efficiently promote calcium mobilization as low concentration. How do you explain it? Inverse agonism? Superagonism? Activation of alternative pathway such as Gq?
Also, the maximum efficacy of the analogs is generally not reported which could be misleading. All the compounds seem to act as partial agonist or weak agonist.
Table 1. Compounds 46, 47, 48. How do you explain the drastic difference of their retention time (11.5 vs 6.7).
Just based on Figure 3, I do not see how the authors can state that both compounds antagonized 85-90% of the agonistic response. Why not calculating the pA2?
Since they had knowledge of the potential molecular interaction between 26RFa(19-26) and its cognate receptor, I do not understand why they did not use more these data to rationally select some analogs right from the beginning and exclude some of them. For instance, they state that compounds 8-10 altered the agonistic activity probably because there is little space available at the bottom of the orthosteric binding site but such information might be available in their in sillico model.
Author Response
Reviewer 1
In the present article, the authors described the impact of point-substitution of phenylalanine residues on the ability of 26RFa(20-26) to mobilize intracellular calcium. This paper is interesting and nicely written except some typos (Greek letter mostly) as well as some missing necessary background. Overall, I am not sure the paper can be publish in its current states sinceI feel that sometimes the authors have a little bit overinterpreted their data/results.
Response:
We thank reviewer-1 for his/her comments on our manuscript. We hope we have answered satisfactorily all the points raised. The amendments in the revised version of the manuscript are marked up using the “track changes” function.
The introduction is informative and nicely written. However, I would have like a little more SAR background in order to better understand right from the beginning the rationale of modifying these phenylalanine residues. Also, a figure, recapitulating the different structure of the compounds cited page 2 line 75-76, would be interesting.
Response:
As requested, we now provide in the Introduction section 3 sentences emphasizing basic SAR data to better introduce the aim of the study (lines 69-74 of the revised manuscript). The compounds mentioned in this section are displayed in the new Figure 2.
Despite referring to a publish article regarding the use of a Ga16-hQRFPR-transfected cell line, I still to not understand why a conventional this particular cell background was needed especially considering the low potency observed with the native ligand.
Response:
We had some difficulties in understanding this comment. Does the reviewer refer to the presence of the G protein alpha16 subunit co-expressed?
The presence of the chimeric Ga16 protein facilitates the process of high throughput screening by circumventing the different coupling of GPCR to distinct intracellular second messenger pathways (New & Wong, 2004, PMID 15285908). We have largely used this type of cell constructions; see for instance: Le Marec et al., 2011, PMID 21623631; Neveu et al., 2012, PMID 22800498, 2014, PMID 24913445; Alim et al., 2018, PMID 30358997).
Also, I was really surprised that no binding affinity is reported. I understand that phenylalanine substitution being at the heart of the article, it was not possible to substitute one of them to introduce of Tyr moiety for subsequent iodination procedure but why not using other strategy to prepare a radioligand. By not investigating this parameter, the author could miss some important information regarding the recognition and activation process.
Response:
We agree with the reviewer comment. We have previously reported binding affinities for 26RFa(20-26) analogues from competition experiments using [125I]-26RFa, i.e. the native full-length sequence, as a tracer (Neveu et al., 2012). Unfortunately, this kind of experiment has been hindered due to the recent removal of the lab into a new building. We are still waiting for authorization from French authorities to hold and experiment with radioactivity in our new restricted areas.
Most if not all of the presented concentration-response curves do not present a clear plateau. One curve, possibly panel D, looks more like a straight line and I do not see how a EC50 can be deduced from it. Also, some of the compounds (difficult to identify which one since the figure is no labelled according to the legend), seem to be better or worse than the reference to efficiently promote calcium mobilization as low concentration. How do you explain it? Inverse agonism? Superagonism? Activation of alternative pathway such as Gq?
Response:
We are sorry to notice that panel labelling and Greek letters have been deleted from our master set. It was beyond our control, and we have corrected these typos (lines 68 and 84, for instance).
As reported in the Materials & Methods section, all the curves have been fitted from Prism 8.0 software using a sigmoidal dose-response fit with variable slope. The curves presented are prototype curves chosen according to their EC50 as close as to the mean EC50 from at least 3 independent experiments. EC50 values have been automatically determined by Prism from raw data. This software is the gold standard and is specifically formatted for data analyses.
Our long-lasting expertise on the cellular system used, prompted us to apply the dose of 10-5 M as the highest; beyond that, responses obtained are not specific. The compounds presented herein are weak agonists, their dose-response curves are shifted to high doses and partially displayed (plateau not observed). However, curve parameters are calculated by the software. A large number of studies report such situation with weak agonists, see for instance pEC50 of compounds 14 in Merlino et al. (2019) PMID 30615452, and EC50 of compound 5 in Le Marec et al. (2011) PMID 21623631.
Regarding the fluorescence of the compounds and reference at low concentrations, we believe that the difference observed cannot be due to the activation of an alternative pathway since we used a system directed to calcium signaling, highly oriented by the Ga16-subunit co-transfection. It may rather reflect experimental variations.
Also, the maximum efficacy of the analogs is generally not reported which could be misleading. All the compounds seem to act as partial agonist or weak agonist.
Response:
Yes, all analogs are weak/partial agonists. To take in consideration the reviewer’s comment that we have "a little bit overinterpreted of our data", we have decided to not provide maximal efficacies calculated by the software.
Table 1. Compounds 46, 47, 48. How do you explain the drastic difference of their retention time (11.5 vs 6.7).
Response:
We were also surprised by the hydrophilic behavior of compound 47 in comparison to compounds 46 and 48. We confirm that all compounds have been chromatographed on the same column and in the same conditions of elution. We hypothesize that phenylalanine side chain of residue 24 may be embedded in the cyclic core, restrained by the bridge between position 22 and 26, and not exposed to the C18 stationary phase of the HPLC column. Indeed, such a steric effect is well documented.
Just based on Figure 3, I do not see how the authors can state that both compounds antagonized 85-90% of the agonistic response. Why not calculating the pA2?
Response:
Once again, the data were analyzed using the Prism software. To answer the referee’s request, we have provided the percentage of antagonism efficiency at the highest dose tested. (Table 2).
Since they had knowledge of the potential molecular interaction between 26RFa(19-26) and its cognate receptor, I do not understand why they did not use more these data to rationally select some analogs right from the beginning and exclude some of them. For instance, they state that compounds 8-10 altered the agonistic activity probably because there is little space available at the bottom of the orthosteric binding site but such information might be available in their in sillico model.
Response:
Some of the compounds presented in the present manuscript were synthesized and evaluated before our docking study. Our objective was to display herein all the molecular diversity of a homogenous library that we have designed around the three phenylalanine residues from our pioneer investigations until now. As usual in all SAR studies, some compounds are rationally designed, while others are more opportunist.
To satisfy the referee’s comment, we now refer to our docking study to discuss the data obtained for compounds 8-10 (line 107 of the revised version).
Reviewer 2 Report
The study by Lefranc et al. describes a detailed structure-activity relationship investigation on the C-terminal heptapeptide of 26RFa, an important but understudied endocrine transmitter. The design of the study is thorough and very detailed, the presentation of the results is clear and straightforward. The authors obviously put a lot of effort in analyzing a large number of compounds. It would be nice to know if some of the newly identified agonists with higher potency than the parent peptide display higher receptor selectivity for QRFPR, for example in comparison to NPFF1 or NPFF2 receptors.
I have only a few minor points regarding formatting and spelling. Besides that the study should be ready for publication.
Minor:
- Introduction line 53: please correct "...which depend, at least in part, on off-target..."
- line 63-64: please correct "...associated with full or partial..."
- Results, lines 93 and 196: please correct, either "participates in", or "contributes to"
- Figure 2: please label panels A-E and maybe add legends into each panel that facilitate identification of compounds/peptides
- Table 1 and 3: I would suggest to move them into Supplementary Material, more important are the data presented in Table 2 and 4
- Table 2: please align formatting of the columns so that the numbers are closer together. The MDPI template for tables makes this difficult, I know from own experience.
- lines 165, 170, 190, 228: Greek letter formatting was lost
- line 202: please correct "very"
Author Response
Reviewer 2
The study by Lefranc et al. describes a detailed structure-activity relationship investigation on the C-terminal heptapeptide of 26RFa, an important but understudied endocrine transmitter. The design of the study is thorough and very detailed, the presentation of the results is clear and straightforward. The authors obviously put a lot of effort in analyzing a large number of compounds. It would be nice to know if some of the newly identified agonists with higher potency than the parent peptide display higher receptor selectivity for QRFPR, for example in comparison to NPFF1 or NPFF2 receptors.
I have only a few minor points regarding formatting and spelling. Besides that the study should be ready for publication.
We thank reviewer-2 for his/her very positive comments on our manuscript. Unfortunately, we do not have in the lab experimental tools to screen and evaluate the specificity of our compounds on NPFF1 and NPFF2 receptors. We think that this type of experiment could be deputed to a CRO only to investigate hit compounds. The amendments in the revision versed of the manuscript are marked up using the “track changes” function.
Minor:
- Introduction line 53: please correct "...which depend, at least in part, on off-target..."
Response:
Corrected
- line 63-64: please correct "...associated with full or partial..."
Response:
Corrected
- Results, lines 93 and 196: please correct, either "participates in", or "contributes to"
Response:
Corrected (twice)
- Figure 2: please label panels A-E and maybe add legends into each panel that facilitate identification of compounds/peptides
Response:
It appears that some characters were altered between our master set and the formatted document used for peer review. We apologize for these inconsistencies.
- Table 1 and 3: I would suggest to move them into Supplementary Material, more important are the data presented in Table 2 and 4
Response:
We prefer to keep in the manuscript these data, which are very informative for a chemist readership.
- Table 2: please align formatting of the columns so that the numbers are closer together. The MDPI template for tables makes this difficult, I know from own experience.
Response:
We prefer to entrust this task to MDPI type-setters, at the risk of doing worse.
- lines 165, 170, 190, 228: Greek letter formatting was lost
Response:
Corrected
- line 202: please correct "very"
Response:
Corrected
Round 2
Reviewer 1 Report
The authors have adequately addressed all my comments.